# Bridging the Gap between Research and the Community: Implementing Physical and Cognitive Interventions to Improve Spontaneous Walking Speed in Older Adults

**DOI:** 10.3390/ijerph20010762

**Published:** 2022-12-31

**Authors:** Kristell Pothier, Navin Kaushal, Tudor Vrinceanu, Maxime Lussier, Nathalie Bailly, Francis Comte, Thien Tuong Minh Vu, Nicolas Berryman, Louis Bherer

**Affiliations:** 1Department of Psychology, PAVeA Laboratory (EA 2114), University of Tours, 37041 Tours, France; 2Research Centre, Institut Universitaire de Gériatrie de Montréal, Montréal, QC H3W 1W5, Canada; 3Department of Health Sciences, School of Health and Human Sciences, Indiana University, Indianapolis, IN 46202, USA; 4Research Centre, Montreal Heart Institute, Montréal, QC H1T 1C8, Canada; 5Department of Medicine, Université de Montréal, Montréal, QC H3C 3J7, Canada; 6Research Centre, Centre Hospitalier de l’Université de Montréal, Montréal, QC H3C 3J7, Canada; 7Département des Sciences de l’Activité Physique, Université du Québec à Montréal, Montréal, QC H2X 1Y4, Canada

**Keywords:** walking speed, community centre, physical interventions, cognitive training, feasibility study

## Abstract

The application of interventions to enhance mobility in ecological settings remain understudied. This study was developed to evaluate the feasibility of training methods in a community centre and to evaluate their impact on mobility outcomes. Fifty-four participants were randomized to one of three 12-week training programs (three times/week): aerobic (AE), gross motor abilities (GMA) or cognitive (COG). Feasibility was evaluated by calculating adherence, feedback from participants and long-term participation. The impact of these interventions on mobility was assessed by comparing pre- and post-program on Timed-up-and-go (TUG) and spontaneous walking speed (SWS) performances. Results showed relatively high rates of adherence (85.1%) and long-term participation (66.7%), along with favorable feedbacks. SWS significantly improved in COG (0.10 ± 0.11 m.s^−1^; *p* = 0.004) and AE (0.06 ± 0.11 m.s^−1^; *p* = 0.017) groups, and TUG performance was maintained in all groups. Results of this feasibility study demonstrated successful implementation of physical and cognitive training programs, encouraging the development of real-world applications.

## 1. Introduction

Mobility is one of the most important components of healthy ageing. Several standardized protocols, such as the Timed-Up and Go test (TUG) or the assessment of Spontaneous Walking Speed (SWS) can reliably predict various health conditions in aging [1,2]. SWS has a powerful clinical relevance since values below 1 m.s^−1^ are associated with a high risk of disability, cognitive impairment, institutionalisation, falls, and/or mortality in older adults [3]. Improving SWS has also been associated with a greater life expectancy: older adults who increased their SWS by 0.10 m.s^−1^ over a year were found to substantially reduce their risk of death during an 8-year follow-up (58% reduction in relative risk and 17.70% reduction in absolute risk of death) [4]. Even smaller improvements in walking speed (Δ > 0.05 m.s^−1^) have been found to translate into meaningful improvements in performance of daily living activities (e.g., stair climbing, walking a block) [5]. Therefore, increasing older adults’ SWS is a critical challenge for sustained healthy aging.

Previous research studies have shown that SWS can be improved through different types of interventions. A systematic review recently showed that being involved in aerobic trainings induced significant gains in older adults’ SWS [6]. For example, a specific five-month progressive aerobic training (four days a week, 15 to 30 min at an intensity between 50% and 70% of the heart rate reserve) has led to a 0.08 m.s^−1^ change in SWS in sedentary older adults aged 65–79 years old [7]. It was also recently suggested that development of gross motor abilities (coordination, agility and/or flexibility) could improve mobility: a previous meta-analysis including eight interventional studies (*n* = 198) showed a 0.09 m.s^−1^ change in SWS in healthy older adults [8]. Studies also suggest that SWS can be improved after cognitive interventions. Literature is now abundant on the strong links between cognitive processes (mainly executive functions and attention) and gait [9,10]. Recent reviews and meta-analyses highlighted the benefits of computerized cognitive training on brain adaptations and mobility outcomes [11,12]. Encouraging results were observed, especially on balance [13] and on SWS, with a 0.13 m.s^−1^ improvement [14] among a sample of sedentary older adults after a three-month intervention focused on executive control. A recent original study also compared the impact of aerobic, gross motor abilities, and cognitive training programs on determinants of mobility performance and revealed similar benefits induced by the three interventions on the TUG scores, despite non-significant changes in SWS [15]. 

While effective in lab-research settings, health benefits of interventions adapted and conducted within the community need to be further studied. Several studies designed to promote physical activity have been conducted at home and did not lead to positive changes in SWS (see 16, for a review). The absence of peer support, the lack of supervision by professionals and suboptimal facilities can represent important obstacles, preventing improvements in key outcomes [16,17,18]. Interestingly, community center-based programs could address these issues and, therefore, represent a unique environment to transfer the knowledge gained in a lab environment to the aging population living in the community. Recently, Marusic et al. (2022) replicated an 8-week lab-based computerized cognitive training in a daily activities center for older adults. They found gait improvements under challenged condition in trained older adults, compared to a wait-list control group [19]. Yet, to the best of our knowledge, no study has evaluated and compared if different physical and cognitive lab-based protocols could be transferred in community centers. Another important challenge for health care professionals is the adherence of older adults to physical activity programs, considering that trained individuals might be able to avoid mobility limitations associated with sedentary behaviours [20].

Therefore, the main objective of this study was to evaluate the feasibility of implementing physical and cognitive interventions, known to improve older adults’ mobility, in a community centre and to evaluate the impact of these programs on mobility performance. The hypothesis is that members of the community center will adhere to aerobic, gross motor abilities and cognitive interventions originally designed in a research environment. This program should allow SWS and TUG improvements, regardless of the intervention.

## 2. Materials and Methods

### 2.1. Study Design

This study aims to translate evidence-based research [15] to clinical practice. This is an open label intervention study with a three-arm design. Participants were assigned using blocked randomization, stratified according to age and sex. After collecting demographic and clinical data, participants were invited to three pre-intervention visits evaluating physical and cognitive outcomes. Participants were then randomized to one of the following three training protocols: Aerobic (AE), Gross Motor Abilities (GMA) or Cognitive (COG) program. After the twelve-week training program, participants completed the same assessments in the same order as pre-tests. Evaluators administering the pre- and post-intervention tests were blind to participants’ assigned interventions and trainers were not involved in the pre- and post-intervention evaluations. No study-related adverse events (e.g., injuries) have been reported in any of the groups.

### 2.2. Participants

Participants were recruited from a local recreational centre for seniors, through advertisements. Ninety-four older adults showed interest in the study and decided to contact the project’s recruitment team. Inclusion criteria included individuals who were: non-smokers, consumed none to a moderate amount of alcohol (≤2 standard alcohol units per day), and who have not undergone any hormone therapy treatment. Participants were excluded if they underwent major surgery or were diagnosed with significant medical illness within the past year, were having contraindications to perform physical activity or limited mobility, thyroid or pituitary glands diseases, neurological disease or early signs of dementia (Mini Mental State Examination [21], MMSE < 26), depression (Geriatric Depression Scale [22], GDS ≥ 11), major uncorrected auditory or visual impairments, and diagnosis of orthopaedic, cardiovascular, respiratory progressive somatic or psychiatric problems within the last six months. A research assistant collected medical history and treatments for each participant and administered the new Physical Activity Readiness Questionnaire (PAR-Q+) [23] to identify possible limitations to performing physical activity. Participants who answered “yes” to, at least, one of the seven questions were not able to participate without approval from the geriatrician from our team. A brief neuropsychological evaluation was also administered to assess global cognitive functioning (MMSE), abstract verbal reasoning (Similarities of the Weschler Adult Intelligence Scale—WAIS III), processing speed (Digit Symbol Coding subtest of the WAIS III), and working memory (Digit Span backward/forward subtests of the WAIS III [24] at baseline.

Of the 94 interested participants, 40 were excluded as per the eligibility criteria. Included participants (*n* = 54) were equally randomized into one of the three intervention groups. Eleven participants were excluded or dropped out from the study (schedule conflicts, personal reasons or dislike of the program) leaving a total of 43 healthy older adults (AE: *n* = 13; GMA: *n* = 14; COG: *n* = 16) in the final sample (see Figure 1). The mean age of participants was 68.14 (SD = 4.93) years. The participants consisted of 65% women, the mean educational level was 13.79 years (SD = 3.07) and the mean BMI was 27.74 (SD = 4.64). 

The study protocol was approved by the Ethics committee and all participants signed the consent form. All participants included in the study were instructed to avoid any change in their daily routines and eating habits.

### 2.3. Intervention

All programs included a total of 36 training sessions (3 sessions weekly—Mondays, Wednesdays, Fridays) for 60 min per session. The AE and GMA groups were supervised by certified kinesiologists employed at the community centre, who were previously trained by our research kinesiologist. The COG group was supervised by two volunteers from the centre, who were already involved in a memory-training program there. They were both trained by the neuropsychologist who designed this intervention arm. Participants were required to attend at least 75% of the training sessions to be included in the final analysis.

#### 2.3.1. Aerobic Intervention

AE training was based on a protocol used in previous studies [15,25] and was adapted to respect the constraints of the community center. It consisted of a low to moderate intensity long interval protocol performed entirely on a spinning-type stationary bike. Each training was made of 6 bouts of 5 min cycling at 60–80 revolutions per minutes. During the first week, the trainers helped the participants find this pace with a metronome. The rate of perceived exertion (RPE) was used to quantify AE training intensity. The 10-point Borg scale [26] was first explained to the participants prior to training and resistance was then self-adjusted on the cycling ergometer to achieve the prescribed RPE for each bout of each session. Prescribed RPE varied between 3/10 and 7/10 RPE. The first and the last 5 min bouts of each session were, respectively the warm-up and the cool-down of the training. The prescribed RPE intensity were 3–4/10 RPE the first month, then 4/10 for the second month, and 4–5/10 for the last month. The main part of the training was made of four long intervals of 5-min, with 3 to 5 min of rest between the intervals and prescribed RPE 5/10 to 7/10. The rest between the intervals started at 5 min for the first week, then went down to 4 min and 3 min on the second and third week, respectively. Then, on the fourth week of the first month, the rest period went back to 5 min between the intervals, but the prescribed RPE increased to 6/10 instead of 5/10. The second and third month of the protocol went in a similar fashion, apart from the last training week which was identical to the before-last week of training: four long intervals of 5 min at 7/10 RPE with 3 min rest period between intervals. After each training bouts (warm-up, intervals, cool down) the trainers asked the participants their RPE to ensure they were training at the prescribed intensity and adjusted the resistance accordingly.

#### 2.3.2. Gross Motor Abilities Intervention

The GMA training was based on a program used in previous studies [15,25]. This training consisted in a set of motor exercises during which aerobic intensity was kept very low. Each one-hour session started with a 10 min walking period at a spontaneous and comfortable speed. Thereafter, participants had to complete different exercises designed to improve coordination, balance and agility. On Mondays, exercises with a focus on locomotion and lower body coordination were prioritized. On Wednesdays, exercises targeting balance were planned whereas on Fridays, hand-eye coordination (i.e., aiming and throwing) was prioritized. Throughout the program, exercises were added combining multiple skills (coordination, agility, balance) to increase the level of difficulty. For instance, participants had to maintain balance on one foot before throwing a ball in a box or to walk sideways while maintaining an object in their hands. Participants did those GMA exercises for approximately 30 min, and then completed another 10 min walking period. Each session was concluded with 5 min of stretching to increase overall body flexibility and some breathing exercises to allow participants to cool down physically and mentally. For all sessions, participants were asked to maintain their normal breathing frequency as much as possible in order to minimize improvements in cardiorespiratory fitness. 

#### 2.3.3. Cognitive Intervention

The COG intervention was performed on an individual computer, through a dedicated web-based computerized neuropsychological battery centered on executive functions [27]. The training, used in a previous study [15], contained three different tasks for participants:-Dual-Task to train divided attention: The DT paradigm involved performing two discrimination tasks alone or concurrently. Participants had to answer as fast as possible, while making as few errors as possible, to one or two stimuli (fruits vs. modes of transport, letters vs. numbers or sounds vs. beeps) appearing in the center of the tablet by pressing on the appropriate button of a digit keyboard with their left and/or right thumbs. Stimuli were presented visually or orally, since each participant had headphones. After two training sessions, participants were asked to prioritize one hand over the other, depending on the trials, in order to increase the level of difficulty.-Stroop task to train inhibition and switching: The digit modified Stroop task consisted of four different conditions. First, in the Reading condition, digits from one to six were presented in small identical groups corresponding to their numerical values (e.g., four copies of the digit “4”) and participants had to press the corresponding digit (“1” to “3” with their left thumb on the keyboard; “4” to “6” with their right thumb on the keyboard) as fast as possible while making as few errors as possible. In the Counting condition, groups of one to six asterisks were presented and the participants had to report how many asterisks were present. In the Inhibition condition, digits were presented in small identical groups and the digits presented were incompatible with the number of digits presented (e.g., five copies of the digit “4”). Participants were asked to count how many digits were presented, and avoid reporting the identity of the digit. Finally, stimuli in the Switching condition were identical to those of the Inhibition condition, except that, for one random trial out of each sequence of five trials, the group of digits was surrounded by a white frame, indicating that, for those trials only, participants had to report the identity instead of the quantity of digit(s). To manipulate the level of difficulty in this task, stimuli and their position on the screen were often changed along the training weeks.-N-back task to train updating: The n-back task is a continuous cognitive task. Stimuli (digits from “1” to “9”) were presented sequentially and participants had to indicate if the current digit matched the one from *n* steps earlier in the sequence. Digits were presented visually, on the screen, and were also heard in each participant’s headphone as a voice spoke each digit as they were displayed. For the present study, the load factor *n* could vary from one to three. Two response buttons were displayed on the right side of the keyboard. The one above was for the response “is the same” and the one bellow was for “is different”. Only the right thumb was used for this task. During the first month, only 1- and 2-back were presented. At the beginning of the second month, 3-back was incorporated and for the third month of training, only 2- and 3-back were administered.

Each training session was composed of approximately 20 min of each task.

### 2.4. Feasibility Measures

The feasibility assessment included participants’ adherence, feedback and long-term participation. These measures have been previously used to assess feasibility of a multidomain intervention study [28,29].

#### 2.4.1. Adherence

Adherence was evaluated by calculating retention rate (i.e., the percentage of the 54 randomized participants who did not drop out throughout the 12-week intervention). 

#### 2.4.2. Participants’ Feedback

Feedback from participants was collected throughout a questionnaire of satisfaction filled out at the end of the intervention. This questionnaire was composed of four questions scored from 1 (very little appreciated) to 5 (very much appreciated). Two questions were about participants’ appreciation of the interventions and trainers (Question 1- “To what degree did you like the program you were part of?”; Question 3- “To what degree did you like your trainer?”). Two other questions were asking how much participants would recommend the interventions and trainers (Question 2- “To what degree would you recommend this program to your family or friends?”; Question 4- “To what degree would you recommend your trainer to your family or friends?”). 

#### 2.4.3. Long-Term Participation

Descriptive statistics about the pursuit of training programs at the community centre, collected nine months after the end of the research study, were also verified. Indeed, this centre added two training programs (AE and GMA interventions) to their list of activities at the end of the research project. Long-term participation was evaluated by calculating the long-term retention rate (i.e., after nine months, the percentage of older adults who completed the study (*n* = 43) and later enrolled in one of the AE or GMA programs).

### 2.5. Mobility Measures 

Gait speed has been assessed prior to randomization and at 12 weeks, by using a 10-m walking test in which participants had to walk at their usual pace for ten meters on a straight line [30]. Timing gates (TC-System, Brower Timing Systems, Draper, UT, USA) were used to eventually calculate walking speed in m.s^−1^ for each participant. Three trials were done and the mean score was kept for analyses. Following previous recommendations, we used a clinical cut-off score of 0.05 m.s^−1^ [4] to assess the clinical effects of these interventions (AE, GMA and COG). 

TUG performance was assessed before and after the 12-week intervention. Participants had to rise from a chair (with armrests), walk three meters on a straight line to reach a cone, turn around the cone, walk back to the chair, and sit down [31]. Participants were asked to walk at their usual gait speed. Three trials were administered. Each trial was timed (in seconds), and the average performance of the three trials was kept for analyses. 

### 2.6. Statistics 

All analyses were performed using SPSS Statistics 28.0. Following descriptive statistics on adherence, participants’ feedback and long-term participation to assess the feasibility of the study, outcomes of the program were assessed by conducting a repeated measures ANOVA. Specifically, Group (AE vs. GMA vs. COG) was placed as the between-subject factor, and Time (pre- and post-tests) was set as the within-subject factor for the SWS and TUG tests. Associated to partial eta squared for the ANOVAs, the magnitude of the observed differences was assessed for each group using Hedges’ g [32]. The magnitude of the effect was considered small (0.2 < ES ≤ 0.5), moderate (0.5 < ES ≤ 0.8), or large (ES > 0.8) following Cohen’s guidelines [33].

## 3. Results

### 3.1. Baseline Data

All baseline neuropsychological scores are presented in Table 1. The three groups did not significantly differ in all these baseline variables. However, differences were found at baseline for SWS as GMA participants were significantly faster than their COG counterparts (SWS_COG_ = 1.28 ± 0.20; SWS_GMA_ = 1.48 ± 0.26; *p* = 0.005).

### 3.2. Feasibility

#### 3.2.1. Adherence

Of the 54 participants enrolled, 46 completed the intervention, representing a 85.1% retention rate. Among the eight dropouts, three occurred during the pre-tests (*n* = 3), and five were during the 12-week intervention. Specifically, one participant was from the AE group who dropped out due to scheduling conflicts, three dropped out from the GMA arm due to personal reasons, and one participant dropped out from the COG arm because the participant did not like the program.

#### 3.2.2. Participants’ Feedback

The answer to Question 1, regarding how much they appreciated the program, obtained a mean score of 1.66/5 (±0.85). When looking at each training group, mean scores were 1.62/5 (±0.77) for AE, 1.54/5 (±0.78) for GMA and 1.80/5 (±1.01) for COG. When asked about their inclination to recommend the intervention (Question 2), participants responded favourably as demonstrated by an overall mean score of 4.37/5 (SD = 0.86). When looking at each training group, mean scores were 4.62/5 (±0.65) for AE, 4.46/5 (±0.66) for GMA and 4.07/5 (±1.10) for COG. Generally, participants did appreciate their trainers with mean scores to Question 3 reaching 4.29/5 (SD = 0.98). Group analyses suggest that AE and COG had higher scores than GMA: 4.46 (±0.97), 4.47 (±0.64) and 3.92 (±1.26), respectively. Mean score to Question 4 was 4.39/5 (SD = 0.86). Scores for each intervention were AE: 4.62 (±0.65), GMA: 4.54 (±0.66) and COG: 4.07 (±1.10). 

#### 3.2.3. Long-Term Participation

Descriptive statistics provided by the community center revealed that 69.23% (9/13 participants) of the AE group and 64.28% (9/14 participants) of the GMA group registered to continue training in their respective programs for nine months after the intervention. No cognitive training was offered at the community centre at the end of the study, due to organizational reasons, but five participants from the COG group (31%) subscribed in AE (*n* = 1) and GMA (*n* = 4) programs after their intervention. No transfer between AE and GMA occurred. 

### 3.3. Mobility Measures 

#### 3.3.1. SWS 

Pre and post walking speed values are presented in Table 2. Repeated-measures ANOVA showed a Time effect (F(1.39) = 8.572, *p* = 0.006, η^2^
*p* = 0.17) but no Group*Time interaction (*p* = 0.105). The overall improvement in SWS across all groups was 0.06 m.s^−1^, 95% CI [0.02, 0.10], SD = 0.13. When looking at each group, results showed a moderate (g = 0.51) and significant (*p* = 0.004) SWS improvement of 0.10 m.s^−1^ (95% CI [0.04, 0.16], SD = 0.11) for COG. The AE group showed a moderate (g = 0.50) and significant (*p* = 0.017) increase of 0.06 m.s^−1^ (95% CI [−0.01, 0.14], SD = 0.13). No SWS change was observed for GMA (0.00 m.s^−1^, 95% CI [−0.07, 0.08], SD = 0.13; g = 0.01). 

#### 3.3.2. TUG 

Pre and post TUG scores are presented in Table 2. Repeated-measures ANOVA showed no significant Time effect (*p* = 0.588) or Group*Time interaction (*p* = 0.326), indicating that participants maintained their TUG performance after the three 12-week interventions. However, when looking at each group, results showed a small (g = −0.23) non-significant (*p* = 0.365) TUG improvement of −0.26 s (95% CI [−0.57, 0.04], SD = 0.51) for AE.

## 4. Discussion

The objective of this study was to evaluate the implementation of different training programs in a community centre, and to assess their impact on mobility, which is a crucial health outcome in older adults. Results showed that all three former laboratory-based interventions were generally well-received by older participants and rather well-implemented in the community centre as they led to an overall meaningful SWS change.

Results showed a high retention rate, with more than 85% of the participants completing the 3-month interventions. A review of interventional studies focused on physical activity showed a mean retention rate of 81% in community-based studies [17]. The slightly higher rate in our study shows a good implementation of these interventions within the community centre. This rate is in accordance with most of participants’ feedback obtained. Three of the four questions participants had to answer at the end of the program obtained high scores. Remarkably, older adults would recommend the three interventions and trainers, which indicates good overall feedback. However, based on one appreciation score, participants did not seem to enjoy the interventions. Nevertheless, given the fact that more than 60% of trained participants decided to keep on doing their respective interventions despite this overall low score, it is highly possible that they could appreciate the benefits of such programs. Similar to taking unpleasant but efficient drugs to heal specific diseases, it seems older adults would start thinking about physical and cognitive interventions as highly important non-pharmacological strategies to delay age-related functional decline and would comply with it. Of important note, the social aspect of training in a familiar community centre could have facilitated compliance with these non-pharmacological interventions and increase positive results. 

Overall, SWS improved in average by 0.06 m.s^−1^ in our healthy population despite the fact that the interventions implemented underwent some adjustments to be in line with the community setting. Prior research was conducted with high-intensity interval aerobic training (HIIT), which has been associated with improvements in cardiorespiratory fitness and functional performances in older adults [25]. However, the implementation of such a HIIT program generally needs medical screening and supervision, which are often unavailable in a community centre. Nevertheless, and importantly, these adjusted interventions led to a SWS improvement that is, overall, slightly over the clinical cut-off of 0.05 m.s^−1^ [5]. Considering that SWS tends to decrease from 12% to 16% per decade after 63 years old [34], maintaining or even increasing such an important vital sign through specific interventions in older adults is of great importance to help prevent further age-related declines. 

This study showed that different types of trainings could induce this improvement. Although the literature is quite abundant on efficient physical training programs that maintain older adults’ mobility [35], the positive impact of cognitive interventions on older adults’ gait has been shown more rarely. Here, results showed that both AE and COG trainings led to a significant clinical gait change (0.06 and 0.10 m.s^−1^, respectively), indicating that different ways seem to exist to improve an important indicator of older adults’ health. Based on the potential energy concept [36], we could hypothesize that AE training directly impacts walking speed by improving cardiovascular fitness. Regarding COG intervention, it is likely that the improvement lied on the strong links and shared neural substrate connecting cognitive processes and gait [9,10]. Nevertheless, the GMA training group did not show any walking improvement although this group specifically trained key motor abilities, including agility and balance. This lack of improvement could be due to a training program that was not intensive enough for a highly functional sample. Indeed, the older participants included here were quite young (m_age_ = 68 yo ± 5), with a high educational level (an average of 13.80 ± 3.10 years of education) and with walking speeds at baseline over normal values (m_women_ = 1.36 m.s^−1^ (±0.20) and m_men_ = 1.41 m.s^−1^ (±0.22) with respective normal values of 1.24 m.s^−1^ and 1.34 m.s^−1^ [37]), representing a sample of fit older adults. Recently, it has been suggested that for such highly fit populations, training prescription should target the development of lower body strength and power [20]. Indeed, the initial mobility status and adaptive potential relationship highlights that high functioning individuals require targeted interventions to eventually enhance mobility outcomes [20]. 

The fact that no significant changes were observed for TUG performance is quite surprising, especially considering that similar interventions implemented in a research center were associated with improvements [15]. This phenomenon could be associated with a ceiling effect as the actual sample of participants had baseline values representing faster TUG performances, which could limit the potential for adaptations. 

Methodological limitations must, however, be highlighted. The literature suggests that improvements of 0.05 and 0.10 m.s^−1^ represent worthwhile clinical changes [5]. Nonetheless, it has to be reminded that the minimal detectable difference is most likely greater than these values [38]. Therefore, capturing these adaptations remains an important challenge for clinicians, especially with high fit populations. From a practical perspective, clinicians are obviously encouraged to standardize as much as possible testing conditions. Regular testing, possible with accessible protocols such as the TUG and the SWS, could also provide an indication of the outcome’s normal variability for each participant. Then, if true improvements are difficult to confirm, especially in fit persons, the attention could be on performance maintenance, which is still crucial from an aging perspective.

Taken together, positive feedback from participants associated with the overall SWS improvement indicate that community centre-based interventions seem to be feasible. While laboratory projects offer control on a high number of variables and allow verifying precise hypotheses about potential mechanisms, the implementation of interventions outside of laboratories constitutes an important ecological research progress. Indeed, it pushes researchers to share the knowledge obtained from these well-controlled laboratory experiments with coaches, caregivers and other actors working closely with a specific population. Moreover, such an approach leads to discussions and adjustments that are necessary to meet the community characteristics. Compared to home-based interventions, the rich social environment and the strong professional support offered by recreational centres have a high potential to increase the efficacy of training programs. Indeed, as Stillman et al. [39] recently highlighted, socioemotional functions (e.g., mood, motivation) could mediate the relationship between physical activity and cognition performances in addition to cellular, molecular and structural brain changes. Here, more than the content of the programs itself, participants could have appreciated social interactions in a familiar environment, both factors that could, at least partially, explained the positive impact of these interventions. Despite different population and outcomes, an Australian study [40] also highlighted the beneficial impact of a community centre-based resistance training on glycemic control in overweight sedentary adults with type 2 diabetes compared to a home-based intervention. Authors highlighted the fact that it is always easier to adhere to a program within a formal group in comparison to being isolated at home. Our study also pointed out the efficiency of peer-led interventions. Indeed, two older volunteers were in charge of the COG intervention, the training program that led to the highest SWS change. This result is in accordance with a recent systematic review showing the efficiency of peer-led programs in aging populations [41] and further supports more ecological research programs.

While encouraging, this feasibility study had some limitations that should be pointed out: (1) Feasibility measures and statistics. This study included three measures (adherence, participants’ feedback and long-term participation) allowing not only for facilitated data collection within the community center but also a scientific assessment of feasibility. Nevertheless, these measures could have been completed to obtain more information, especially on participants who dropped out throughout the 12-week intervention (related to the adherence measure), and on potential non-compliance in participants who enrolled in one of the AE or GMA program at the end of the research study (related to the long-term participation measure). These additional measures would have resulted in new statistics (e.g., intention-to-treat analyses) that would have improved the interpretation of the data. (2) The small sample size. This feasibility study suffers from a lack of sample size calculation, and the final analysis included a relatively small sample of participants, preventing us from generalizing our results. However, to the best of our knowledge, this is the first study aiming at transferring both lab-based physical and cognitive interventions within a community center. Positive results found here may encourage the development of larger randomized controlled studies. (3) The influence of the lifestyle background of participants on the results. Inclusion and exclusion criteria used in this study were intended to homogenize the sample as much as possible and thus reduce the risk of having participants with very different lifestyle habits. In addition, all participants included in the study were instructed to avoid any change in their daily routines and eating habits. Nevertheless, we cannot totally exclude the risk that certain unmeasured individual (e.g., personality traits, motivation, perceived self-efficacy, spontaneous physical activity) or environmental (e.g., marital status, living environment) variables known to influence exercise participation may have influenced the results.

## 5. Conclusions

Effectively translating training approaches that prevent functional declines into the community is essentially the ultimate goal of clinical research. Yet, testing real-world application has received limited attention in the literature. Specifically, as demonstrated by high retention scores, positive feedback and improvements in SWS, the present study demonstrated successful implementation of physical and cognitive training programs.

## Figures and Tables

**Figure 1 ijerph-20-00762-f001:**
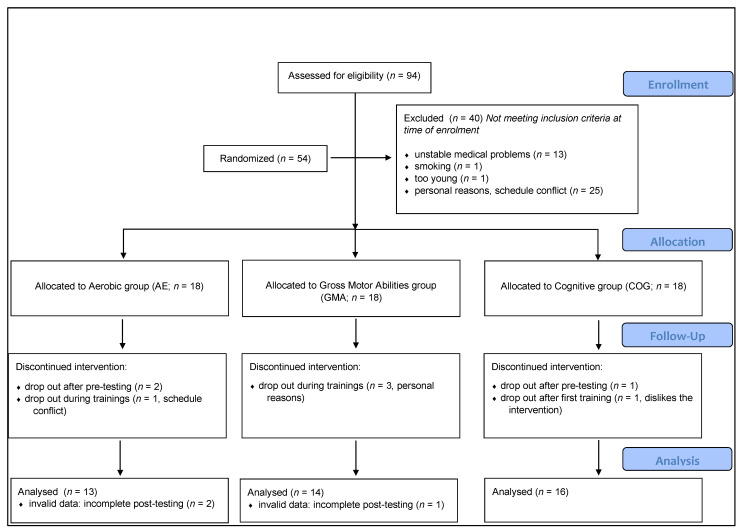
Randomization process of the study.

**Table 1 ijerph-20-00762-t001:** Baseline characteristics.

	AE	GMA	COG	F or X2	*p*
** *n* **	13	14	16	-	-
**Age**	67 (4.69)	68.93 (5.90)	68.19 (4.52)	0.497	0.61
**Sex** (women/men)	8/5	10/4	10/6	0.367	0.83
**Education** (number of years)	13.85 (3.36)	12.57 (2.73)	14.81 (2.89)	2.101	0.14
**BMI** (kg/m^−2^)	27.16 (4.02)	26.79 (3.83)	29.04 (5.64)	1.020	0.37
**GDS**	2.46 (3.12)	2.00 (2.11)	3.69 (4.06)	1.089	0.35
**MMSE** (/30)	28.54 (1.56)	28.50 (1.65)	28.75 (1.18)	0.128	0.88
**Digit span** (forward + backward)	16.92 (4.07)	17.00 (3.84)	16.69 (4.39)	0.024	0.98
**Similarities**	20.08 (4.90)	20.50 (5.98)	22.00 (3.52)	0.646	0.53
**DSST**	64.15 (9.60)	56.86 (16.93)	62.94 (1.75)	1.295	0.28

Abbreviations: BMI: Body Mass Index; GDS: Geriatric Depression Scale; MMSE: Mini Mental State Examination; DSST: Digit Symbol Substitution Test; AE: Aerobic training; GMA: Gross Motor Abilities training; COG: Cognitive training.

**Table 2 ijerph-20-00762-t002:** Mobility performances per intervention group.

	Baseline Mean (SD)	12 Weeks Mean (SD)	Mean Change (SD) [95% CI]	Hedges’ g
**Spontaneous Walking Speed (m.s^−1^)**				
**AE**	1.39 (0.09)	1.45 (0.14)	0.06 * (0.13) [−0.01, 0.14]	0.500
**GMA**	1.48 (0.26) ^#^	1.48 (0.21)	0.00 (0.13) [−0.07, 0.08]	0.009
**COG**	1.28 (0.20)	1.38 (0.16)	0.10 * (0.11) [0.04, 0.16]	0.505
**Timed-Up and Go Test (s)**				
**AE**	7.88 (1.10)	7.72 (0.88)	−0.26 (0.51) [−0.57, 0.05]	−0.226
**GMA**	7.70 (2.16)	8.02 (1.50)	0.27 (1.46) [−0.61, 1.16]	0.121
**COG**	7.99 (1.32)	8.24 (1.38)	0.25 (0.92) [−0.24, 0.73]	0.174

Abbreviations: AE: Aerobic training; GMA: Gross Motor Abilities training; COG: Cognitive training; SD: Standard Deviation; CI: Confidence Interval. *: pre et post change, *p* < 0.05; ^#^: significantly different from the COG group at baseline (*p* = 0.005).

## Data Availability

The datasets used and/or analysed during the current study are available from the corresponding author on reasonable request.

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
