# Peer review of "Bridging the Gap between Research and the Community: Implementing Physical and Cognitive Interventions to Improve Spontaneous Walking Speed in Older Adults"

_ijerph, 2022, doi:10.3390/ijerph20010762_

Round 1
Reviewer 1 Report
The authors submitted a manuscript to the IJERPH journal in which they presented the effects of 12-week training program in older adults. They tested a 3x/week aerobic, gross motor abilities and cognitive intervention on walking speed, while being interested in adherence to the intervention. The results show overall high adherence and improved walking speed in all three interventional groups. While the manuscript Is well written, there are several issues I point below. My major concerns are related to statistical method selected, a limited number of outcomes presented in this paper and lack of related studies that are not mentioned in this manuscript. Retention effects are unclear and would increase an overall quality of the paper as well as practical importance. Final sample size included in analysis is low and there is no sample size calculation.
In the introduction authors introduce studies that relate to physical and mental training, and mention that there are no studies that tested lab-based protocols to be implemented in community settings. I believe that literature review should be presented in more details, including all necessary studies. I would suggest several modifications. First, expand and restructure the second paragraph in introduction. I believe there should be better overview of previous studies (possibly high quality randomized controlled trials) segmented in three subsections that you address: aerobic training, gross motor skill training and cognitive training where details are presented. Below I provide an example for cognitive training to improve mobility outcomes. The same should be for other two categories.
In row 52-58 you miss a large amount of evidence, which was performed in the lab of dr. Verghese. Namely, study from 2010, which was performed in parallel with Li et al 2010; Meta-analysis from 2018, which summarized the effects of cognitive-based interventions to improve mobility; finally, in 2021 there is a large-scale study published on 383 participants. Study from 2022 (Does Cognitive Training Improve Mobility, Enhance Cognition, and Promote Neural Activation?) tried to replicate the lab-based protocol from Verghese et al 2010 in Center for daily activities for the elderly, where active and independently living older adults performed computerized cognitive training for 2 months.
In the methods authors lack further justification of selected interventions. While their description is detailed, it is not clear on which basis (previous studies) the protocols were chosen.
In rows 241-247 it is not clear whether participants circulated or turned back to the initial point for three times.
Due to small sample size and dropouts, it is not clear why repeated measures ANOVA was preferred and not intention to treat analysis.
How did the authors control for baseline differences in gait speed between groups in main analysis?
Again, are the outcome measures limited to SWS and TUG only?
I personally think that the limitation section should be expanded with future directions described in more details.
Minor corrections:
R48: remove double spaces twice, same in R159
Author Response
RESPONSES TO REVIEWER 1
We sincerely thank you for your review. We have carefully read each of your comments and tried to address them as best we could.
Reviewer 1: “In the introduction authors introduce studies that relate to physical and mental training, and mention that there are no studies that tested lab-based protocols to be implemented in community settings. I believe that literature review should be presented in more details, including all necessary studies. I would suggest several modifications. First, expand and restructure the second paragraph in introduction. I believe there should be better overview of previous studies (possibly high quality randomized controlled trials) segmented in three subsections that you address : aerobic training, gross motor skill training and cognitive training where details are presented. Below I provide an example for cognitive training to improve mobility outcomes. The same should be for other two categories.”.
Answer: Following your advice, we have expanded the Introduction and added, in the second paragraph and for each subsection, systematic reviews or meta-analyses (see Reference 6 [Levin et al., 2017], Reference 8 [Hortobagyi et al., 2015], and Reference 12 [Marusic et al., 2018] in the revised manuscript) to emphasize the strong links between the three interventions and mobility improvement. We deliberately let or added details in Henderson et al. (2017, Reference 7), in Hortobagyi et al. (2015, Reference 8), and in Pothier et al. (2018a, Reference 14) studies to highlight specific examples on SWS change after aerobic, gross motor and cognitive trainings.
We also thank you for your examples on cognitive training. We have added two of the given references (Marusic et al., 2018, Reference 12, and Marusic et al., 2022, Reference 19) in the revised Introduction, which improves the manuscript.
R1: “In the methods authors lack further justification of selected interventions. While their description is detailed, it is not clear on which basis (previous studies) the protocols were chosen”.
Answer: In the beginning of the Methods section (l.90), we wrote that the “study aims to translate evidence-based research (15) to clinical practice”. This statement was written to invite readers to refer to the three lab-based interventions used in Pothier et al. (2021; Reference 15). To ensure more clarity, we have added a sentence related to this rationale to the description of each selected intervention in the revised manuscript (see l. 146-147, l.169, and l.188).
R1: “In rows 241-247 it is not clear whether participants circulated or turned back to the initial point for three times.”
Answer: The TUG test asks participants to walk from a chair to a cone three meters away. After a turn back (around the cone), participants walk back to the chair to sit down. This test was done three times (three trials). Each trial was timed, and the average performance was retained for the analyses.
We modified this paragraph in the revised manuscript (l.259-264) to ensure more clarity.
R1: “Due to small sample size and dropouts, it is not clear why repeated measures ANOVA was preferred and not intention to treat analysis.”
Answer: Thank you for this relevant comment. Intention-to-treat (ITT) analyses are indeed known to be the gold standard for analyzing the results of clinical trials. Nevertheless, for this feasibility study aiming at transferring lab-based interventions within a community center, we deliberately chose to conduct the same statistical analyses done in the previous study from which the interventions were drawn (hence, facilitating the comparisons).
In addition, ITT analyses can only be carried out fully if there is a complete and full set of data (Armijo-Olivo et al., Physical Therapy Reviews, 2009), including details on drop-outs and non-compliance, which was not the case in this feasibility study unfortunately. We do agree though that ITT analyses would have improved the interpretation of the data and decided to expand the limitation section accordingly (see l.441-449).
R1: “How did the authors control for baseline differences in gait speed between groups in main analysis?”
Answer: Thank you for your careful reading. Indeed, participants from the GMA group differently differed from the COG group in spontaneous walking speed (SWS) values at baseline. Considering this, we thought of performing our analyses on relative (instead of absolute) SWS pre-to-post changes. Nevertheless, we wanted to be able to compare our data with recognized clinical thresholds (Perera et al., 2006). Since these meaningful SWS change are based on absolute pre-to-post values, we deliberately chose to perform our analyses on these data.
The purpose of this implementation study is not to evaluate the effectiveness of the interventions on mobility, but to provide a description of the effects observed in a real environment. We are aware that this clinical objective may suffer from certain methodological biases. Therefore, and considering your remarks, we have expanded our limitation section in the hope of nuancing the interpretation of our results while insisting on the importance of developing research in ecological settings.
R1: “are the outcome measures limited to SWS and TUG only?”
Answer: This study aims to translate evidence-based research to clinical practice. While it could represent a limitation, we specifically wanted to find, in this feasibility study, the right balance between controlling a maximum of variables and being in an ecological environment. We therefore chose to limit the use of certain tools and measure mobility through SWS and TUG only. In future studies, it would be interesting though to extend these findings to more accurate and precise outcomes (using gaitmat or electric walk mats for instance) or to add other physical parameters (e.g., balance, strength).
R1: “I personally think that the limitation section should be expanded with future directions described in more details.”
Answer: Following another reviewer’s advice and yours, we have expanded the limitation section in the revised Discussion (see l.440-463). We have specifically taken into account your relevant comments on retention effects and small final sample size.
R1: “Minor corrections: R48: remove double spaces twice, same in R159”
Answer: Minor corrections (removing double spaces) have been done.
Reviewer 2 Report
Some minor suggestions to be addressed but otherwise an interesting and excellent paper.

Author Response
RESPONSES TO REVIEWER 2
Thank you very much for your review. Above are our answers to the five questions/comments related to the Methods section.
Reviewer 2: “P3 l85: type of randomisation?”
Answer : We used a blocked randomization, stratified according to age and sex. This information has been added to the revised manuscript l.92.
R2: “P3 l86: Would there not be a case for at least concealing allocation for stats? And were the outcome assessors blinded?”
Answer: Regarding masking, participants were aware of the three intervention groups available and knew only after the pre-intervention evaluations which of the three groups they were assigned to. They were aware of the main purpose of the study (“to evaluate the feasibility of implementing physical and cognitive interventions, known to improve older adults’ mobility, in a community center”). Evaluators administering the pre- and post-intervention tests were blind to participants’ assigned interventions and trainers were not involved in the pre- and post-intervention evaluations. This missing information has been added to the revised manuscript l.97-99.
R2: “P3 l98: Rationale for this exclusion not clear”
Answer: For more clarity, we modified the sentence “major uncorrected perceptive limitations” into “major uncorrected auditory or visual impairments” (l.111 in the revised manuscript) .
R2: “P3 l99: Vague especially given the population of older adults”
Answer: The sentence “presence of a somatic or known progressive psychiatric pathology” was vague indeed. We were more specifically looking for a diagnosis of orthopaedic, cardiovascular, respiratory progressive somatic or psychiatric problems within the last six months. This has been added in the revised manuscript l.112-113.
R2: “P3 l109: Bearing in mind the issues concerning MMSE regarding language and education, you may need to talk about the participants in these terms at some point”
Answer: Cognitive scores are often impacted by education or language level. In this study, we used MMSE only to ensure a relatively consistent global cognitive level in our sample (all three groups did not significantly differ in MMSE scores). Participants included in this study were, by the way, highly functional (this has been raised in the discussion l.390-394).
Reviewer 3 Report
With regard to manuscript: Bridging the Gap between Research and the Community: Implementing Physical and Cognitive Interventions to Improve Spontaneous Walking Speed in Older Adults, for consideration in Int. J. Environ. Res. Public Health. This is a very interesting manuscript addressing interventions (aerobic, gross motor abilities or cognitive) to enhance mobility in ecological settings. The paper will contribute to knowledge and is worthy of publication. There are few points to be improved.
• The novelty of the study could be more highlighted (in introduction).
• In abstract “Overall, SWS significantly improved (0.06±0.13m.s−1; p=0.009) in all groups”. Please revise this since no SWS change was observed for GMA.
• A limitation of this study is the lack of physiological measures (for example heart frequency) that could validate the aerobic training
• The three groups did not significantly differ in all these baseline variables. This was very right and proper. Congratulations for this excellent study.
• The explanation of the statistics is adequate. I only ask the authors to carefully consider the possibility of using Eta-squared (η2) for evaluating the magnitude of the observed differences by ANOVA.
• Didactics would improve with the inclusion a figure in discussion (some scheme drawn by the authors) explaining their findings.
• Would have the lifestyle background (physical activity at homes “spontaneous physical activity” - SPA) of participants any influence on their findings? I recommend the inclusion of a discussion about this. I would like this point of view to be more in-depth.
Author Response
RESPONSES TO REVIEWER 3
Thank you very much for your review. Above are our answers to the six raised questions/comments.
Reviewer 3: “The novelty of the study could be more highlighted (in introduction).”
Answer: Following another Reviewer’s comment and yours, we have expanded and restructured the Introduction. We also took your specific comment into account and hope now that the novelty of the study is more highlighted in the revised manuscript (see the second and third paragraphs of the revised manuscript).
R3: “In abstract “Overall, SWS significantly improved (0.06±0.13m.s−1; p=0.009) in all groups”. Please revise this since no SWS change was observed for GMA.”
Answer: The abstract has been modified accordingly.
R3: “A limitation of this study is the lack of physiological measures (for example heart frequency) that could validate the aerobic training”
Answer: This study aims to translate evidence-based research to clinical practice. While it could represent a limitation, we specifically wanted to find, in this feasibility study, the right balance between controlling a maximum of variables and being in an ecological environment. We therefore chose to limit the use of certain tools and measures. Nevertheless, all the three trainings used here have demonstrated their efficacy previously (see Pothier et al., Exp. Geront., 2021). More specifically regarding the aerobic training, previous results showed a significant pre-to-post improvement in peak oxygen uptake (VO2 peak, a physiological measure of cardiorespiratory fitness) after the same 12-week aerobic training (ΔPost-Pre, in ml.kg-1.min-1, with 95% CI = 1.932 [.618 to 3.245]; see Pothier et al., Exp. Geront., 2021).
R3: “The explanation of the statistics is adequate. I only ask the authors to carefully consider the possibility of using Eta-squared (η2) for evaluating the magnitude of the observed differences by ANOVA.”
Answer: Following the Reviewer’s advice, we have added partial eta squared when appropriate (see l.271 and l.321)
R3: “Didactics would improve with the inclusion a figure in discussion (some scheme drawn by the authors) explaining their findings.”
Answer: We do agree that a figure often improves the understanding and retention of the key messages of a paper. As written in the conclusions of the manuscript, the key message of this feasibility study would be to encourage efficient implementation of physical and cognitive training programs into the community through testing real-world application in order to prevent functional declines in aging. While interesting, the idea of drawing a scheme representing this message seems difficult to implement here.
R3: “Would have the lifestyle background (physical activity at homes “spontaneous physical activity” - SPA) of participants any influence on their findings? I recommend the inclusion of a discussion about this. I would like this point of view to be more in-depth.”
Answer: Thank you for this relevant comment. The influence of unmeasured variables on the results of intervention studies is a fundamental concern. Inclusion and exclusion criteria used in this study were intended to homogenize the sample as much as possible and thus reduce the risk of having participants with very different lifestyle habits. In addition, all participants included in the study were instructed to avoid any change in their daily routines and eating habits (see l.134-135). Nevertheless, we cannot totally exclude the risk that certain unmeasured individual (e.g., personality traits, motivation, perceived self-efficacy, spontaneous physical activity) or environmental (e.g., marital status, living environment) variables known to influence exercise participation may have influenced the results. This important limit has been added to the Discussion in the revised manuscript (l.455-463).